# Vancomycin-Resistant Enterococci: Screening Efficacy and the Risk of Bloodstream Infections in a Specialized Healthcare Setting

**DOI:** 10.3390/antibiotics14030304

**Published:** 2025-03-16

**Authors:** Assunta Navarra, Stefania Cicalini, Silvia D’Arezzo, Francesca Pica, Marina Selleri, Carla Nisii, Carolina Venditti, Angela Cannas, Antonio Mazzarelli, Antonella Vulcano, Stefania Carrara, Donatella Vincenti, Barbara Bartolini, Paolo Giacomini, Maria Grazia Bocci, Carla Fontana

**Affiliations:** 1Clinical Epidemiology Unit, National Institute for Infectious Diseases “Lazzaro Spallanzani”, IRCCS, 00149 Rome, Italy; assunta.navarra@inmi.it; 2Systemic and Immune Depression-Associated Infections Unit, National Institute for Infectious Diseases “Lazzaro Spallanzani”, IRCCS, 00149 Rome, Italy; 3Microbiology and Biobank Unit, National Institute for Infectious Diseases “Lazzaro Spallanzani”, IRCCS, 00149 Rome, Italy; silvia.darezzo@inmi.it (S.D.); marina.selleri@inmi.it (M.S.); carla.nisii@inmi.it (C.N.); carolina.venditti@inmi.it (C.V.); angela.cannas@inmi.it (A.C.); antonio.mazzarelli@inmi.it (A.M.); antonella.vulcano@inmi.it (A.V.); stefania.carrara@inmi.it (S.C.); donatella.vincenti@inmi.it (D.V.); barbara.bartolini@inmi.it (B.B.); carla.fontana@inmi.it (C.F.); 4Department of Experimental Medicine, University of Rome “Tor Vergata”, 00133 Rome, Italy; pica@uniroma2.it; 5Health Direction, Hospital Information Service, National Institute for Infectious Diseases “Lazzaro Spallanzani”, IRCCS, 00149 Rome, Italy; paolo.giacomini@inmi.it; 6Intensive Care Unit, National Institute for Infectious Diseases “Lazzaro Spallanzani”, IRCCS, 00149 Rome, Italy; mariagrazia.bocci@inmi.it

**Keywords:** vancomycin-resistant enterococci (VRE), cultural tests, molecular tests, bloodstream infections (BSIs), antimicrobial surveillance protocols

## Abstract

**Background**: Vancomycin-resistant enterococci (VRE) rectal colonization represents a critical risk factor for subsequent bloodstream infections (BSIs), posing a serious concern in healthcare settings. This study aims to investigate the association between the presence of VRE in rectal swabs and the occurrence of BSIs, highlighting the challenges of rapid detection and patient care implications in an infectious disease hospital setting. **Methods**: We performed a retrospective analysis of cultural rectal swab screening and molecular assays (MAs) for VRE detection between January 2020 and December 2023. All adult patients admitted with at least one rectal swab screening performed during hospitalization were included. All blood cultures that yielded VRE were identified, and the first Enterococcus-positive blood sample for each patient with at least one prior rectal swab per year was analyzed. **Results**: The results showed a 15.4% positivity rate for VRE in cultural screening, predominantly *Enterococcus faecium*. MA showed a higher prevalence of 49.4%, with a significant discordance between MA rectal swab screening and cultural testing. Patients with VRE intestinal colonization by *E. faecium* were significantly more likely to develop *E. faecium* BSI, with a risk ratio of 9.78 (*p* < 0.001). **Conclusions:** The study identified a strong correlation between VRE rectal colonization and the risk of developing BSI, emphasizing the need for effective screening and infection control measures. The results support the inclusion of molecular testing in VRE detection protocols and highlight the importance of ongoing surveillance for antimicrobial resistance.

## 1. Introduction

Enterococci are Gram-positive and facultatively anaerobic cocci, usually arranged in short chains (two to eight cells) and cause severe infections, including bloodstream infections (BSIs), infective endocarditis, urinary tract infections, intra-abdominal infections, and, rarely, meningitis [1]. Enterococci, including *Enterococcus faecalis* and *Enterococcus faecium*, have inherent resistance to various antibiotics. *E. faecalis* resists β-lactams (ampicillin, amoxicillin, imipenem), aminoglycosides (gentamicin, streptomycin), and fluoroquinolones (ciprofloxacin, levofloxacin). It exhibits low-level resistance to penicillins due to alterations in penicillin-binding proteins (PBPs) and to aminoglycosides due to poor uptake and the *lin*B gene. *E. faecium* is particularly resistant to β-lactams, aminoglycosides, and glycopeptides (vancomycin, teicoplanin), mainly due to the *van*A and *van*B genes. It also shows low-level resistance to penicillins, aminoglycosides, and fluoroquinolones through similar mechanisms as *E. faecalis*. Moreover, enterococci can also acquire antibiotic resistance, which can be transferred to other bacteria via mobile genetic elements, plasmids, and transposons and can interact with each other and with the bacterial chromosome to form complex mobile structures [2,3]. Vancomycin-resistant enterococci (VRE) are a global challenge. In recent decades, the epidemiology of VRE has changed markedly, and an increasing trend in the resistance of enterococci to vancomycin, particularly *E. faecium*, has been observed in many countries [4]. The increase in the number of VRE infections in hospitals is of great concern. Data from the Central Asian and European Surveillance of Antimicrobial Resistance (CAESAR) Network show a significant increase in *E. faecium* invasive isolates (+59%) across twelve European countries [4]. In the European Union and European Economic Area (EU/EEA) (excluding the United Kingdom), the estimated incidence of invasive *E. faecium* isolates increased (+25.9%) from 8.5 cases per 100,000 population in 2019 to 10.7 cases per 100,000 population in 2023 [5]. In Italy, the number of recorded VRE cases continues to increase. The latest National Antimicrobial Resistance Surveillance System (AR-ISS) report indicated a percentage of VRE *faecium* equal to 32.5% in 2023 [6]. This increasing trend highlights the need for close monitoring to better understand the epidemiology of VRE, at both the community and hospital levels. It has been hypothesized that the coronavirus disease 2019 (COVID-19) pandemic influenced healthcare delivery and the overall composition of hospital patients, potentially affecting the patterns of enterococcal BSIs, particularly in intensive care units (ICUs) [7]. A high frequency of enterococcal BSIs was also reported in patients with severe COVID-19 admitted to intensive care facilities in a retrospective observational study conducted at a reference center for infectious diseases in northern Italy in 2020 [8]. Moreover, a study population conducted in northwestern Italy reported an increasing trend of VRE positivity in rectal swab samples collected at hospital admission during the COVID-19 pandemic [9].

However, in addition to the effect of the COVID-19 pandemic, other factors to explain the trend observed for enterococcal BSIs should be explored. Enterococci are common colonizers of the gastrointestinal tract in healthy individuals, and resistant enterococci densely colonize the gut following antibiotic treatment [10].

Indeed, the use of antibiotics in hospitals increases the occurrence and spread of drug-resistant enterococci, making these microorganisms one of the most common causes of hospital-acquired infections, particularly in the ICU setting [11].

In a study of hematologic patients, vancomycin administration and altered bowel habits were the only independent risk factors for nosocomial VRE colonization, although the authors reported that VRE colonization and infections caused by VRE did not influence 30- or 90-day mortality [12].

The risk of developing an infection caused by VRE varies, depending on the type of patients and their hospital setting [13,14,15]. The risk of developing an infection caused by VRE varies depending on the type of patient and the hospital setting [13,14,15]. Recently, in a study assessing the risk of BSI following rectal colonization by MDROs in three hospitals between 2019 and 2022 in Greece, a country with a generally high percentage of reported antimicrobial resistance similar to that found in Italy, previous VRE rectal colonization significantly increased the risk of subsequent VRE BSI by 2.5 (1.5–4.2) [16].

As infections caused by VRE are more likely in patients with rectal VRE colonization than in those without rectal VRE colonization, improving target screening strategies for the early identification of patients at risk and limiting the nosocomial spread of VRE and consequent invasive infections caused by VRE are needed [12].

The use of rectal swabs for VRE screening is widespread and aims to detect carriers, prevent transmission, and identify patients who would benefit from empiric VRE treatment in cases of enterococcal bacteremia [17,18].

At present, there are two primary methods for the testing of rectal swabs for VRE: the traditional culture technique and molecular assay (MA) [11,14,17,19,20]. Molecular testing is usually fast and accurate and allows healthcare professionals to manage patients quickly and appropriately, but it is expensive and may detect the target gene *van*A/B from nonviable VRE isolates, potentially leading to discordant results [17,21].

In our facility, VRE surveillance by rectal screening is performed at admission for patients with risk factors for VRE colonization.

The objectives of the study were (i) to evaluate the effectiveness of rectal screening using MAs and (ii) to determine the association between VRE colonization detected by rectal swabs and the likelihood of developing a BSI caused by VRE.

## 2. Results

### 2.1. Patient Population

The National Institute for Infectious Diseases “Lazzaro Spallanzani” is a referral hospital for infectious diseases. It is a scientific institute for hospitalization and care, unique in Italy for infectious diseases, with about 160 beds.

During the study period, 5599 patients were hospitalized, accounting for a total of 6053 hospital admissions (5341 patients were hospitalized once, 258 were hospitalized twice or more). Among the 6053 hospital admissions, 3793 (3793/6053; 62.7%) were due to COVID-19.

On each admission, patients were screened for VRE rectal colonization according to their risk factors at presentation (Figure 1). Specifically, patients were screened by MA, and positive MA results were confirmed by culture. Patients admitted before 2021 were tested using culture only, as the MA assay had not yet become available.

The results of the screening tests are given in the following sections.

### 2.2. VRE Rectal Colonization by Culture Tests

The total number of VRE cultural screenings evaluated was 11,593, with a median of one screening for each admission (IQR 1–2; range 1–30). Overall, 15.4% (1783/11,593) of the rectal screening tests yielded a positive result for VRE, 98.7% (1760/1783) yielded a positive result for *E. faecium*, and 0.3% (23/1783) yielded a positive result for *E. faecalis*. Among all patients admitted to the hospital, 888 (14.7%) had VRE rectal colonization, 870 had *E. faecium* rectal colonization (98%), 11 had *E. faecalis* rectal colonization (1.2%), and 7 (0.8%) had both *E. faecalis* and *E. faecium* rectal colonization. No other enterococci other *E. faecalis* and *E. faecium* were identified. The median time from admission to the first cultural swab yielding VRE was 1 day (IQR: 0–10; range 0–135). The median time to result was 56.6 (IQR = 16.9) h.

### 2.3. VRE Rectal Colonization by MA

A total of 2101 MA rectal screening test results were evaluated, of which 1037 (49.4%) indicated positivity; specifically, 586 (56.5%) yielded a positive result for vanA, 380 (36.7%) yielded a positive result for vanB, and 71 (6.8%) yielded a positive result for both vanA and vanB. A median of one MA screening test per admission (IQR: 1–2; range from 1 to 16) was performed. All 1037 of the positive MA rectal samples were cultured; 521 (50.2%) yielded a positive result for VRE, and 516 (49.8%) yielded a negative result for VRE; in particular, 21.8% of the vanA-positive tests yielded a negative result for VRE, whereas 97.9% of the vanB-positive tests yielded a negative result. MA provided results earlier than the culture test, with a median time to result of 18.9 (IQR = 5.8) h.

### 2.4. VRE Bloodstream Infections

We analyzed admissions with positive blood cultures for VRE who had previously had a rectal swab to assess the association between rectal colonization with VRE and the development of BSI. A total of 42 (0.7%) VRE-positive blood culture samples were identified, including 37 (37/42; 88%) with *E. faecium* and 5 (12%) with *E. faecalis*. The median length of hospital stay before the identification of VRE in blood cultures was 15 days (IQR 4–28; range 1–148). The median time from the first rectal culture positive for VRE to a blood culture positive for VRE was 11 days (IQR 5–42; range 0–127 days). None of the patients with a positive rectal culture for VRE were treated with an antibiotic effective against VRE before VRE BSI. Thirty-four BSIs caused by VRE occurred in COVID-19 patients, with an incidence of 3.84 (95% CI 2.74–5.37) cases per 10,000 person-days compared with 1.58 cases (95% CI 0.79–3.15) in those without COVID-19 (IRR 2.44, 95% CI 1.13–5.26; *p* = 0.024). Among the 37 patients with BSIs due to *E. faecium*, 23 (62.2%) had at least one positive VRE rectal culture screening, whereas 14 did not (Table 1). In contrast, among the five patients with BSIs caused by *E. faecalis*, only one had a VRE-positive rectal culture screening. Overall, 24 of the 886 (2.71%) VRE-positive rectal swabs and 18 of the 5167 (0.35%) VRE-negative swabs resulted in BSIs caused by VRE. Patients with VRE-positive rectal swabs were significantly more likely to develop a BSI than those with VRE-negative swabs (RR 7.78; 95% CI 4.24–14.27, *p* < 0.001) (Table 1(a)). Patients with positive VRE rectal screening results due to *E. faecium* infection had an increased risk of developing a BSI caused by VRE *faecium* (RR 9.78; 95% CI 5.05–18.94, *p* < 0.001) (Table 1(b)). Since MA assays were introduced in 2021, there have been 11 VRE BSIs with an associated MA assay, of which 7 tested positive. No statistical analysis was performed due to the small sample size.

### 2.5. Antimicrobial Susceptibility of the Isolates

Table 2 shows the changes in the antimicrobial susceptibility of the isolates over time. When antimicrobial resistance trends in both *E. faecalis* and *E. faecium* were examined, several notable patterns emerged. The resistance of *E. faecalis* to ampicillin increased slightly from 0% in 2020 to 3.23% in 2023, whereas for *E. faecium*, it increased from 73.85% in 2020 to 100% in 2023. Clindamycin was consistently 100% resistant in both species across all years. The resistance of *E. faecalis* to gentamycin-HL peaked at 46% in 2021 but decreased to 38.71% by 2023, whereas for *E. faecium*, it increased from 55.38% in 2020 to 73.33% in 2023. The resistance of *E. faecalis* to imipenem fluctuated, peaking at 12% in 2021 before decreasing to 6.45% in 2023. For *E. faecium*, it increased from 75.38% in 2020 to 100% in 2023. The resistance of *E. faecalis* to linezolid was 3.23% in 2023, whereas for *E. faecium*, it peaked at 5.13% in 2022 before decreasing to 0% in 2023. The resistance of *E. faecalis* to teicoplanin increased from 1.37% in 2020 to 6.45% in 2023, whereas for *E. faecium*, it varied, peaking at 41% in 2022 and decreasing to 33.33% in 2023. Tigecycline resistance was minimal in both species, with the resistance of *E. faecalis* peaking at 1.79% in 2022 and that of *E. faecium* peaking at 0% in 2023. Both species were consistently 100% resistant to trimethoprim–sulfamethoxazole and tobramycin. Finally, the resistance of *E. faecalis* to vancomycin increased from 1.37% in 2020 to 6.45% in 2023, whereas for *E. faecium*, it varied, peaking at 38.46% in 2022 before decreasing to 33.33% in 2023. Table 3 presents the detailed MIC distribution for daptomycin. Notably, the *E. faecalis* isolates predominantly presented a MIC of 2, whereas the MIC of *E. faecium* ranged between 2 and 4.

## 3. Discussion

The recent increase in infections caused by VRE in hospital settings is highly significant [5,22,23,24]. VRE is responsible for a substantial number of infections in healthcare settings and is associated with high mortality rates, particularly in patients with BSIs [25]. Moreover, VRE infections are more likely in patients with hospital-acquired VRE rectal colonization than those without [12]. In this context, rectal screening for VRE may be considered a crucial strategy to prevent the spread of these resistant bacteria within healthcare environments and to counteract their increasing prevalence effectively. Several studies have highlighted the need to monitor patients for VRE colonization, strongly recommending that rectal screening be prioritized, particularly in critically ill patients [12,26,27]. Rectal swabs are commonly used for VRE screening to detect carriers, prevent transmission, and identify those patients who may need empiric treatment for enterococcal bacteremia.

In our study, we investigated the association among VRE rectal colonization, molecular diagnostic tools, and BSI in a specialized healthcare setting. We observed a 15.4% VRE positivity rate, with many positive cases due to *E. faecium* (98.7%) and a small proportion due to *E. faecalis* (0.3%), suggesting a significant presence of *E. faecium* in the hospital environment, which is of concern given its association with severe hospital-acquired infections. Additionally, in our study, patients with VRE rectal colonization, particularly by *E. faecium*, were significantly more likely to develop VRE bacteremia (risk ratio for VRE *faecium* colonization 9.78; *p* < 0.001), with a median time to infection of 11 days. These findings are consistent with those reported in the literature. Indeed, in a systematic review of 16 studies mostly focused on high-risk populations (e.g., organ transplant patients, patients with malignancies) or high-risk settings (e.g., ICUs), the cumulative incidence of infection after VRE colonization was 8% at 30 days, with a median time to infection of 17 days [28], and the risk of BSI after colonization was 0.9%. Moreover, in a study conducted in a hematology unit in Italy, VRE colonization represented a significant risk factor for VRE infections, with a high prevalence of bacteremia [12]. Finally, in a recent retrospective population-wide cohort study, the risk of infection after detection of VRE colonization was 2.8% within a median of 37 (IQR 11–119) days [29].

Another important point is the value of culture-based rectal screening compared to MA. In our study, MA exhibited a higher positivity rate of 49.4%. However, only 50.2% of these positive samples confirmed positive results on the culture. This discrepancy raises concerns about the clinical significance of PCR detection of VRE in culture-negative patients. In particular, the presence of the *van*B gene did not appear to be associated with culture results, as most *van*B-positive tests yielded negative cultures. Indeed, the presence of the *van*B gene in other intestinal anaerobic bacteria can confound the detection of VRE; consistently, non-enterococcal anaerobic bacteria, such as *Clostridium* spp., *Eggerthella lenta*, and *Ruminococcus* spp., may contribute to a high *van*B false-positive rate [30,31]. Therefore, distinguishing between *van*B-positive anaerobic bacteria and *van*B-positive enterococci is critical to accurately assess the presence and risk of VRE colonization and VRE infections. This distinction should be communicated to clinicians to avoid the misinterpretation of MA results or otherwise not reported to clinicians. Additionally, the emergence of vancomycin-variable enterococci (VVE), which are *van*A-positive but vancomycin-susceptible, further complicates this scenario, because they can be responsible for inconsistent results between MA and culture screening. Although a study by Paule et al. showed that a multiplex PCR assay using *van*A and *van*B primers was more sensitive than culture on selective media for the detection of VRE in rectal swabs (*p* < 0.001) [21], and other studies show that PCR provides results at least two days earlier than traditional culture methods, which is critical for the effective management of nosocomial VRE infections [21,30], our results indicate that culture still has the greatest clinical relevance. In addition, the costs of PCR assays can be relatively high, so the choice of detection method should be based on the specific needs and resources available in the laboratory [32]. Nevertheless, it is worthwhile to improve targeted screening strategies to identify patients at risk and limit the nosocomial spread of VRE and the consequent invasive infections caused by VRE [12].

Another finding of our study is the significantly increasing trend in the antimicrobial resistance of *E. faecalis* and *E. faecium*, which is consistent with the results of recent studies conducted in Italy and Korea [33,34]. We observed that by 2023, the rate of ampicillin resistance in *E. faecium* had reached 100% and clindamycin resistance to gentamicin-high level and imipenem also increased significantly in *E. faecium*. High resistance rates were also observed for trimethoprim–sulfamethoxazole and tobramycin. In addition, the resistance of *E. faecalis* to linezolid, although minimal, was found to be increased in 2023. Interestingly, we observed a predominant distribution for daptomycin for *E. faecalis* isolates at MIC 2, whereas the distribution for *E. faecium* fell between MICs 2 and 4. This finding indicates a relatively lower level of resistance compared to other strains. Daptomycin is currently not approved by the EMA for treating enterococcal infections. The EUCAST lists daptomycin breakpoints for *Enterococcus* species as “IE”—insufficient evidence—and daptomycin is not recommended for therapy [35,36]. However, it is increasingly being used to treat enterococcal infections, particularly those caused by VRE isolates [35,37]. The MIC value we observed may support the effective treatment of infections caused by *E. faecalis* with daptomycin, whereas the MIC distribution observed for *E. faecium* isolates suggests a greater variability in susceptibility. These findings highlight the critical need for accurate MIC testing to guide appropriate treatment strategies.

Our study has some limitations; the retrospective design, small sample size, and single-center setting may limit the generalization of our findings to other healthcare settings with different patient populations, infection prevention and control practices, and antimicrobial resistance patterns. In addition, we did not evaluate patients with previous hospitalization outside our setting or the treatment received previously, which could have influenced VRE colonization and infection rates. However, based on the findings of this study, several areas for future research could be suggested. First, research should focus on understanding the clinical implications of MA-positive but culture-negative results and developing guidelines for their interpretation. Second, prospective studies are needed to validate the association between VRE colonization and the risk of BSIs in different healthcare settings. Finally, studies should investigate the cost-effectiveness of incorporating molecular testing into routine VRE screening protocols and its impact on patient outcomes and healthcare resource utilization.

In conclusion, our study revealed a worrying prevalence of VRE, particularly *E. faecium*, in a specialized hospital setting, highlighting the importance of VRE screening of admitted patients to predict the risk of developing BSIs caused by VRE; it provides important insights for implementing active screening in infection control protocols and antimicrobial resistance surveillance.

## 4. Materials and Methods

The study was conducted at the National Institute for Infectious Diseases L. Spallanzani, a research and reference care hospital for patients with infectious diseases with approximately 160 beds, including 17 ICU beds and approximately 3000 discharged patients per year. In our hospital, microbiological screening for VRE is performed by a rectal swab within the first 48 h of admission in all patients with one of the following characteristics, according to our hospital protocol: (i) colonization or infection with VRE in the previous 90 days; (ii) admission to a healthcare facility, including territorial facilities for elderly individuals and rehabilitation facilities, in the previous 6 months (Figure 1). In addition, patients admitted to the ICU and all transplant patients or those awaiting transplantation undergo rectal screening for VRE on admission and weekly during their hospital stay. Repeated rectal swab screening for all other patients during hospitalization is required only if contact occurs after admission or in cases of suspected or confirmed transmission clusters. All adult patients (≥18 years of age) admitted to our hospital with at least one rectal swab screening for VRE performed during their hospital stay were included in the study. Samples for VRE rectal screening and blood cultures (BCs) yielding enterococci processed by the microbiology laboratory at our hospital between January 2020 and December 2023 were identified by reviewing the microbiology computerized database (LIM: Wlab Themix Italia; Rome, Italy. Version 25.3.b1). In 2020, VRE screening relied solely on traditional culture methods; in 2021, a rectal MA to detect vancomycin resistance targets (*van*A/B) was introduced. No additional analysis was performed if the rectal swab was negative according to the MA method. If the swab was positive for one or more resistance markers (*van*A/*van*B) according to the MA method, the microbiology laboratory proceeded with culture. The time to result (time from sample delivery to the final result) for the culture and MA methods was recorded. We examined all blood cultures yielding enterococci during the study period by reviewing the laboratory middleware EpiCenter version 7.5C (Becton Dickinson; Sparks, MD, USA). The original laboratory data used in this study were extracted by excluding duplicates, which means the first *Enterococcus*-positive samples for each patient per year were included, ensuring that each patient had at least one prior rectal swab. Finally, microbiological data were combined with information extracted from our hospital’s electronic medical records, encompassing admission date and COVID-19 infection status. According to our local protocol, contact isolation precautions are implemented for all VRE-infected or VRE-colonized patients.

### 4.1. Microbiological Studies

Rectal swabs were collected for MA using eSwabs (Copan; Brescia, Italy) and were then processed using the Allplex™ Entero-DR Assay (Seegene Inc., Seoul, Republic of Korea). The samples were vortexed and loaded in a Nimbus Workstation (Hamilton Robotics, Reno, Nevada) with a Universal Extraction Kit (Seegene). DNA extraction was carried out with 200 µL of the primary sample, and the DNA was eluted in a volume of 100 µL. Five microliters of extracted DNA was mixed with 20 µL of master mix, and real-time PCR was performed using a CFX96 system (Bio-Rad, Hercules, CA, USA). All procedures were performed according to the manufacturer’s instructions. Test results were interpreted automatically and presented using the Seegene Viewer software Version 3.0 (Seegene). Samples that were positive for the *van*A and/or *van*B gene were cultured on Liofilchem^®^ Chromatic media (Liofilchem, Roseto degli Abruzzi, Italy) for microorganism identification (ID) and antimicrobial susceptibility (AST) determination. The blood cultures were processed for up to 5 days by a continuous monitoring blood culture system (BACTEC™ FX system; Becton Dickinson; Sparks Glencoe, MD, USA) until they became positive or negative. Specifically, blood samples were collected in Bactec Plus Aerobic/F and Bactec PLUS Anaerobic/F vials (Becton Dickinson) delivered to the laboratory as soon as possible and incubated in the Bactec FX system. When positive, BC was promptly processed using WASP automation (Copan; Brescia Italy) to prepare Gram-stained slides and subcultures. The latter were performed on chocolate agar, MacConkey agar, Chapman agar and Sabouraud–dextrose agar (Thermo Fisher Scientific, Basingstoke, UK). The agar plates were incubated at 35 °C under anaerobic (blood agar plate) and microaerobic conditions. Growing colonies were identified using the MALDI TOF Syrius System (Bruker Daltonics, Bremen, Germany) and MBT Compass software version 4.2. Antimicrobial susceptibility testing (AST) of the isolates was subsequently performed via Phoenix panels run on the Phoenix system PMIC^®^ 96 (Becton Dickinson). Susceptibility patterns were interpreted according to the recommendations of the European Committee on Antimicrobial Susceptibility Testing versions 10.0, 11.0, 12.0, and 13.0 published in 2020, 2021, 2022 and 2023, respectively [36,38,39,40].

### 4.2. Statistical Analysis

Continuous variables are presented as medians with interquartile ranges (IQRs) and minimum–maximum ranges, whereas categorical variables are presented as frequency counts and percentages. The incidence of BSIs during the hospital stay was assessed according to COVID-19 status using the incidence rate ratio (IRR) with a 95% CI. To evaluate the risk of developing BSIs, culture swabs that were positive for VRE were compared to those that were negative for VRE. The risk ratio (RR) was determined with a 95% CI. Only rectal swabs collected before blood culture positivity were included in this analysis. A *p*-value less than 0.05 was considered to indicate statistical significance. Statistical analysis was performed using Stata (Stata Corp. 2021. Stata Statistical Software: Release 17. Stata Corp LLC, College Station, TX, USA).

## Figures and Tables

**Figure 1 antibiotics-14-00304-f001:**
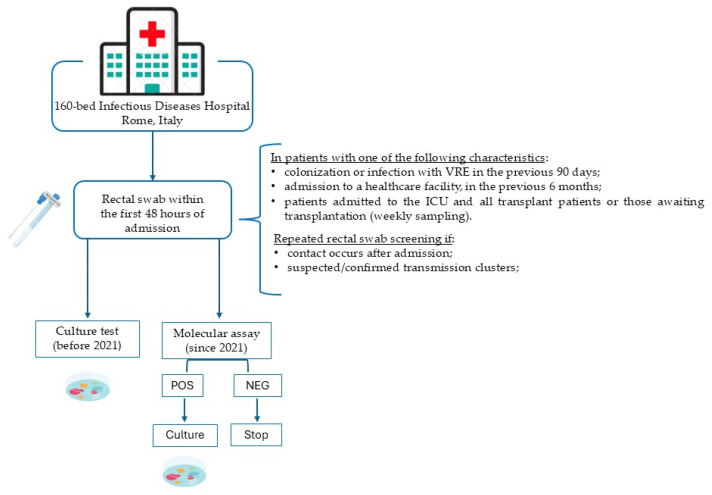
Hospital information and indications for VRE screening (see Section 4 for details).

**Table 1 antibiotics-14-00304-t001:** Risk of BSI-VRE based on rectal swab culture results, (a) overall; (b) for BSI-VRE *faecium* and rectal swab cultures yielding only *E. faecium*.

**(a)**					
**BSI-VRE, N = 42**					
	**Total**	**Positive** **n (%)**	**Negative** **n (%)**	**RR (95% CI)**	** *p* **
**Cultural rectal swabs, N = 6053**					
Positive (*E. faecium* and/or *E. faecalis*)	886	24 (2.7)	862 (97.3)	7.78 (4.24–14.27)	
Negative	5167	18 (0.3)	5149 (99.7)	Reference	<0.001
**(b)**					
**BSI-VRE *faecium*, N = 23**					
	**Total**	**Positive** **n (%)**	**Negative** **n (%)**	**RR (95% CI)**	** *p* **
**Cultural rectal swabs, N = 6030**					
Positive (only *E. faecium*)	867	23 (2.7)	844 (97.3)	9.78 (5.05–18.94)	
Negative	5163	14 (0.3)	5149 (99.7)	Reference	<0.001

VRE: vancomycin-resistant Enterococcus; BSI: bloodstream infection; RR: risk ratio; CI: confidence interval.

**Table 2 antibiotics-14-00304-t002:** Antimicrobial resistance of *E. faecalis* and *E. faecium* isolates from blood cultures (2020–2023).

% of Resistance											
Species	Year	Amp	Clin	Gen-HL	Imi	Lin	Tei	Tig	Tri-Sul	Van	Tob
*E. faecalis*	2020	0.00%	100.00%	31.51%	4.11%	0.00%	1.37%	0.00%	100.00%	1.37%	100.00%
	2021	0.00%	100.00%	46.00%	12.00%	0.00%	2.00%	0.00%	100.00%	4.00%	100.00%
	2022	1.79%	100.00%	32.10%	7.10%	0.00%	0.00%	1.79%	100.00%	1.79%	100.00%
	2023	3.23%	100.00%	38.71%	6.45%	3.23%	6.45%	0.00%	100.00%	6.45%	100.00%
*E. faecium*	2020	73.85%	100.00%	55.38%	75.38%	1.54%	30.77%	1.67%	100.00%	30.77%	100.00%
	2021	92.16%	100.00%	74.51%	96.00%	3.92%	19.60%	0.00%	100.00%	15.69%	100.00%
	2022	84.60%	100.00%	64.10%	92.30%	5.13%	41.00%	0.00%	100.00%	38.46%	100.00%
	2023	100.00%	100.00%	73.33%	10.,00%	0.00%	33.33%	0.00%	100.00%	33.33%	100.00%

Amp: Ampicillin; Clin: Clindamycin; Gen-HL: Gentamycin high level; Imi: Imipenem; Lin: Linezolid; Tei: Teicoplanin; Tig: Tigecycline; Tri-Sul: Trimethoprim–sulfamethoxazole; Van: Vancomycin, Tob: Tobramycin.

**Table 3 antibiotics-14-00304-t003:** Daptomycin MIC distribution of *E. faecalis* and *E. faecium* from 2020 to 2023.

		MIC mg/L				
	Year	0.5	1	2	4	>4
*E. faecalis*	2020	0.0%	30.0%	51.3%	4.1%	2.7%
	2021	9.0%	3.6%	58.2%	9.0%	1.8%
	2022	2.7%	55.5%	22.0%	19.4%	0.0%
	2023	0.0%	3.2%	74.2%	9.6%	0.0%
*E. faecium*	2020	0.0%	6.1%	40.8%	40.8%	12.2%
	2021	39.5%	6.2%	12.3%	39.5%	2.5%
	2022	3.3%	6.6%	0.1%	0.7%	0.1%
	2023	0.0%	0.6%	33.3%	0.4%	0.2%

## Data Availability

The data can be found in the Excel database, created ad hoc and archived at the authors’ institution (INMI L. Spallanzani IRCCS, Rome, Italy).

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
