# Peer review of "Vancomycin-Resistant Enterococci: Screening Efficacy and the Risk of Bloodstream Infections in a Specialized Healthcare Setting"

_antibiotics, 2025, doi:10.3390/antibiotics14030304_

Round 1

Reviewer 1 Report

Comments and Suggestions for Authors

This manuscript by Navarra et al. investigates the correlation between vancomycin-resistant enterococci (VRE) rectal colonization, molecular diagnostic tools, and bloodstream infections (BSIs) in a specialized healthcare setting. The study provides important epidemiological insights essential for infection control and antimicrobial resistance surveillance. The large sample size collected over four years enhances the study’s statistical power and allows for a comprehensive analysis of screening methods, risk factors, and resistance trends.

However, the results section is overly descriptive, and the writing lacks structure, making it difficult for readers to grasp the key findings and their implications. Simply presenting data is not sufficient—the authors should guide the reader through their findings and clearly explain their interpretations and recommendations. Additionally, the discussion requires significant improvement, particularly in integrating key findings into a cohesive, well-structured narrative. I strongly recommend a major revision to enhance clarity, depth, and organization.

Major Comments:

  1. Clarify Correspondence Between MA and Culture Results
    • Your rectal colonization culture tests evaluated 11,593 clinical samples, while MA rectal screening assessed 2,101 samples.
    • Do you have corresponding culture test results for these 2,101 MA-screened samples?
    • In lines 130–132, the statement:

“Among the 1,037 positive MA rectal screening tests, 521 (50.2%) yielded a positive result for VRE, and 516 (49.8%) yielded a negative result for VRE; in particular, 21.8% of the vanA-positive tests yielded a negative result for VRE, whereas 97.9% of the vanB-positive tests yielded a negative result.”
How was VRE positivity or negativity confirmed? Was it based on culture, or another method?

  1. Justify the Results in Lines 146–153
    • What do these results indicate? The discussion lacks an explanation of the clinical significance.
    • Antibiotic treatment consideration:
      • Among 886 VRE-positive rectal swabs, only 24 (2.71%) resulted in BSIs.
      • Are patients with VRE-positive rectal swabs given prophylactic antibiotic treatment to prevent progression to BSI?
    • Statistical consistency:
      • Can the same statistical analysis be performed for MA results as for rectal swab screening?
    • Correlation clarity:
      • What is the precise statistical correlation between rectal VRE positivity and BSIs caused by VRE?
  2. Revise Discussion on Line 276–277
    • The phrase “interesting” is not a sufficient discussion—state why the result is significant.
    • What is your key interpretation here? Avoid vague language and provide clear implications of the finding.
  3. Reorganize and Structure the Discussion
    • The discussion currently consists of too many short, fragmented paragraphs—this disrupts the logical flow.
    • Condense the discussion into four to five structured paragraphs.
  4. Correlation Between Rectal Screening and VRE BSI
    • Discuss whether rectal screening for VRE could be used as a predictive tool for VRE BSI.
    • Should rectal screening be standard for identifying potential BSI risk? What are the clinical implications?
  5. Formatting Issues in Table 2
    • Use dot symbols (.) instead of commas (,) in numerical values.
  6. Typographical Error in Table 2
    • The column label should be "Species", not "Specie".
  7. Clarify CAESAR Abbreviation in Line 61
    • Define CAESAR for readers unfamiliar with this reference.
  8. Introduce Patient Sample Source Earlier (Before Lines 111–113)
    • Provide hospital information and patient selection criteria at the beginning of the Results section. This prevents readers from having to refer back to the Materials and Methods section, improving readability.

Author Response

Reviewer 1

This manuscript by Navarra et al. investigates the correlation between vancomycin-resistant enterococci (VRE) rectal colonization, molecular diagnostic tools, and bloodstream infections (BSIs) in a specialized healthcare setting. The study provides important epidemiological insights essential for infection control and antimicrobial resistance surveillance. The large sample size collected over four years enhances the study’s statistical power and allows for a comprehensive analysis of screening methods, risk factors, and resistance trends.

However, the results section is overly descriptive, and the writing lack’s structure, making it difficult for readers to grasp the key findings and their implications. Simply presenting data is not sufficient, the authors should guide the reader through their findings and clearly explain their interpretations and recommendations. Additionally, the discussion requires significant improvement, particularly in integrating key findings into a cohesive, well-structured narrative. I strongly recommend a major revision to enhance clarity, depth, and organization.

We greatly appreciate your suggestions. We have revised the results section. We have tried to take the reader through our findings in a more narrative way. The interpretations of the results and recommendations are discussed in the discussion section. The discussion has been improved and completely revised according to your suggestions.

Major Comments:

1_Clarify Correspondence Between MA and Culture Results

  • Your rectal colonization culture tests evaluated 11,593 clinical samples, while MA rectal screening assessed 2,101 samples.

Yes, we confirm

  • Do you have corresponding culture test results for these 2,101 MA-screened samples?

In section 2.3 we reported the result of correspondence between MA assay and culture screening as follows “In particular, among 1037 positive MA only 521 (50.2%) yielded a positive result for VRE”

  • In lines 130–132, the statement: “Among the 1037 positive MA rectal screening tests, 521 (50.2%) yielded a positive result for VRE, and 516 (49.8%) yielded a negative result for VRE; in particular, 21.8% of the vanA-positive tests yielded a negative result for VRE, whereas 97.9% of the vanB-positive tests yielded a negative result

 How was VRE positivity or negativity confirmed?

As already stated in the Methods section “No additional analysis was performed if the rectal swab was negative according to the MA method. If the swab was positive for one or more resistance markers (vanA/vanB) according to the MA method, the microbiology laboratory proceeded with culture”. (lines 340-343 of the revised version). Thus, only MA-positive samples were confirmed by cultures. However, we added a figure (Figure 1) to clarify the workflow.

  • Was it based on culture, or another method?

Rectal screening was considered positive based on the culture method. To better clarify we added the sentence “On each admission, patients were screened for VRE rectal colonization according to their risk factors at presentation (Figure 1). Specifically, patients were screened by MA, and positive MA results were confirmed by culture. Patients admitted before 2021 were tested by culture only as MA assay was not available” in the Results section (lines 132-136 of the revised version). We have also added Figure 1 to make the rectal screening workflow easier to understand.

.

2_Justify the Results in Lines 146–153

  • What do these results indicate? The discussion lacks an explanation of clinical significance.

We completely revised the discussion section in the hope that it is now clearer.

  • Antibiotic treatment consideration: Among 886 VRE-positive rectal swabs, only 24 (2.71%) resulted in BSIs. Are patients with VRE-positive rectal swabs given prophylactic antibiotic treatment to prevent progression to BSI?

No, none of the patients underwent prophylaxis. We added a sentence “None of the patients with a positive rectal culture for VRE were treated with an antibiotic effective against VRE before VRE BSI” to clarify this point (lines 168-169 of the revised version).

  • Statistical consistency: Can the same statistical analysis be performed for MA results as for rectal swab screening?

Statistical analysis on MA could not be performed due to the small sample size. We added the sentence “Since MA assays were introduced in 2021, there have been 11 VRE BSIs with an associated MA assay, of which 7 tested positive. No statistical analysis was performed due to the small sample size” (lines 168-169 of the revised version).

  • Correlation clarity:
      • What is the precise statistical correlation between rectal VRE positivity and BSIs caused by VRE?

We have evaluated the association between positive VRE culture tests and VRE BSI. In this regard, we have better described the result, and we have revised Table 2 into a more readable version (see new discussion and Table 2 of the revised version)

3_Revise Discussion on Line 276–277

  • The phrase “interesting” is not a sufficient discussion—state why the result is significant.
  • What is your key interpretation here? Avoid vague language and provide clear implications of the finding.

Thank you for your valuable comment. We have completely revised the discussion section.

4_Reorganize and Structure the Discussion

  • The discussion currently consists of too many short, fragmented paragraphs—this disrupts the logical flow.
  • Condense the discussion into four to five structured paragraphs.

Following your suggestions, we have completely revised the discussion section.

5_Correlation Between Rectal Screening and VRE BSI

  • Discuss whether rectal screening for VRE could be used as a predictive tool for VRE BSI.
  • Should rectal screening be standard for identifying potential BSI risk? What are the clinical implications?

We have completely revised the discussion section, and added these implications. We have removed section “b” from Table 1 as its results were not discussed in the text and did not add, in our opinion, anything relevant to the results.  Section “c” of the table is now renamed “b”

6_Formatting Issues in Table 2

  • Use dot symbols (.) instead of commas (,) in numerical values.

Done.

7_Typographical Error in Table 2

    • The column label should be "Species", not "Specie".

Done

8_Clarify CAESAR Abbreviation in Line 61

    • Define CAESAR for readers unfamiliar with this reference.

Done

9_Introduce Patient Sample Source Earlier (Before Lines 111–113)

    • Provide hospital information and patient selection criteria at the beginning of the Results section. This prevents readers from having to refer back to the Materials and Methods section, improving readability.

Done. We have included the hospital information in the methods and results sections, and we have also added Figure 1. (lines 325-332 of the revised version).

Reviewer 2 Report

Comments and Suggestions for Authors

In this study author investigate the association between the presence of VRE in the rectal swabs and the occurrence of BSIs, highlighting the challenges of rapid detection and patient care implications in a infectious disease hospital setting. In the following section author can find the relevant suggestions.

  1. The introduction could be improved by including the resistance and occurrence of BSI from the rectal VRE infections.
  2. Please include the location of the hospital in the method section.

Author Response

Comments and Suggestions for Authors

In this study, authors investigate the association between the presence of VRE in the rectal swabs and the occurrence of BSIs, highlighting the challenges of rapid detection and patient care implications in an infectious disease hospital setting. In the following section, the author can find the relevant suggestions.

  1. The introduction could be improved by including the resistance and occurrence of BSI from the rectal VRE infections.

We improved the introduction by including occurrence of BSI from the rectal VRE infections adding the sentence “Recently, in a study assessing the risk of BSI following rectal colonization by MDROs in three hospitals between 2019 and 2022 in Greece, a country with a generally higher percentage of reported AMR similar to that in Italy, previous VRE rectal colonization significantly increased the risk of subsequent VRE BSI by 2.5 (1.5-4.2) [Karakosta P, Eur J Clin Microbiol Infect Dis 2025]” (lines 99-103 of the revised version).

We added the reference: Karakosta, P., Meletis, G., Kousouli, E. et al. Rectal colonization with multidrug-resistant organisms and risk for bloodstream infection among high-risk Greek patients. Eur J Clin Microbiol Infect Dis 44, 437–442 (2025). Consequently, we have renumbered the references.

  1. Please include the location of the hospital in the method section.

The location has been included in Methods and the Results.

Reviewer 3 Report

Comments and Suggestions for Authors

This study provides valuable insights into the risks associated with VRE rectal colonization and the subsequent development of bloodstream infections. The findings strongly support the implementation of active screening in infection control protocols. However, the manuscript would benefit from a few minor corrections:

  • Throughout the article, the genus and species names of bacteria (e.g., Enterococcus, Enterococcus faecium, E. faecium) should be italicized. The full name should be provided upon first mention, while subsequent mentions may use the abbreviated genus name (e.g., E. faecalis).
  • The term “enterococci” should not be italicized, as it is a colloquial name.
  • Lines 52-54 – Please revise this section, as not all enterococci are naturally resistant to all beta-lactams, aminoglycosides, or fluoroquinolones. Avoid generalizations that might mislead readers (please refer to EUCAST Expert rules and expected phenotypes).
  • Line 60 – ECDC? Was this meant to be a citation? If so, please provide the reference.
  • Lines 92-94 – Please provide a source for this statement. If these are the authors’ own interpretations, this section should be moved to the discussion or conclusions.
  • Lines 115-124 – What about other Enterococcus species? Were they not identified at all? If they were, why were they not considered in the study?
  • Lines 309-314 – For better readability, I suggest presenting the VRE screening indications in a table. Additionally, please include information on which recommendations were followed and why these particular criteria were chosen.

Author Response

Reviewer 3

Comments and Suggestions for Authors

This study provides valuable insights into the risks associated with VRE rectal colonization and the subsequent development of bloodstream infections. The findings strongly support the implementation of active screening in infection control protocols. However, the manuscript would benefit from a few minor corrections:

  • Throughout the article, the genus and species names of bacteria (e.g., EnterococcusEnterococcus faeciumE. faecium) should be italicized. The full name should be provided upon first mention, while subsequent mentions may use the abbreviated genus name (e.g., E. faecalis)

Done

  • The term “enterococci” should not be italicized, as it is a colloquial name.

Done.

  • Lines 52-54 – Please revise this section, as not all enterococci are naturally resistant to all beta-lactams, aminoglycosides, or fluoroquinolones. Avoid generalizations that might mislead readers (please refer to EUCAST Expert rules and expected phenotypes).

Thank you for your valuable comment. We have corrected and detailed the resistances, primarily for E. faecium and E. faecalis, adding the following sentence” Enterococci, including E. faecalis and E. faecium, have inherent resistance to various antibiotics.   E. faecalis resists β-lactams (ampicillin, amoxicillin, imipenem), aminoglycosides (gentamicin, streptomycin), and fluoroquinolones (ciprofloxacin, levofloxacin). It exhibits low-level resistance to penicillins due to alterations in penicillin-binding proteins (PBPs) and to aminoglycosides due to poor uptake and the linB gene. E. faecium is particularly resistant to β-lactams, aminoglycosides, and glycopeptides (vancomycin, teicoplanin), mainly due to the vanA and vanB genes. It also shows low-level resistance to penicillins, aminoglycosides, and fluoroquinolones through similar mechanisms as E. faecalis”. (see lines 53-60 of the revised version)

  • Line 60 – ECDC? Was this meant to be a citation? If so, please provide the reference.

Sorry, our mistake; it was reference 4.

  • Lines 92-94 – Please provide a source for this statement. If these are the authors’ own interpretations, this section should be moved to the discussion or conclusions.

Done.

  • Lines 115-124 – What about other Enterococcus species? Were they not identified at all? If they were, why were they not considered in the study?

All the Enterococci were identified, with the majority being Enterococcus faecium and Enterococcus faecalis. Rarely, isolates of Enterococcus gallinarum and Enterococcus casseliflavus/flavescens, which carry the vanC gene, were not considered significant because this gene is in the chromosome and does not raise concerns regarding hospital transmission.

However, we have added the sentence “No other enterococci, other E. faecalis and E. faecium were identified. (lines 146-147 of the revised version)

  • Lines 309-314 – For better readability, I suggest presenting the VRE screening indications in a table. Additionally, please include information on which recommendations were followed and why these particular criteria were chosen.

We have added a figure (Figure 1) to clarify the VRE screening indications. Moreover, we have added in the Methods section the sentence “according to hospital protocol” (see lines 329-330 of the revised version)

Round 2

Reviewer 1 Report

Comments and Suggestions for Authors

The quality of the manuscript has been substantially improved. I would suggest to accept and proofread before publication. Thank you.